# Energy Efficiency of Variable Rate Fertilizer Application in Coffee Production in Brazil

Graciele Angnes [1], Maurício Martello [2], Gustavo Di Chiacchio Faulin [3], José Paulo Molin [2] and Thiago Libório Romanelli [1,*]

1   Laboratory of Systemic Management and Sustainability, Department of Biosystems Engineering, Luiz de Queiroz College of Agriculture, University of São Paulo, Piracicaba 13418-900, SP, Brazil; graciangnes@gmail.com

2   Laboratory of Precision Agriculture, Department of Biosystems Engineering, Luiz de Queiroz College of Agriculture, University of São Paulo, Piracicaba 13418-900, SP, Brazil; mauriciomartello@usp.br (M.M.); jpmolin@usp.br (J.P.M.)

3   Fatec Shunji Nishumura, Pompeia 17580-000, SP, Brazil; gustavo.faulin@fatec.sp.gov.br

*   Correspondence: romanelli@usp.br; Tel.: +55-19-3447-8523

**Abstract:** Coffee is a crop of great relevance in socioeconomic terms for Brazilian agribusiness, which is the world's largest producer in cultivated areas. The implementation of precision agriculture in the coffee culture has provided countless benefits to its development, which over the years has been cultivated in the same area. However, there is a lack of studies that address the impact of the application of variable-rates inputs in soil on the energy efficiency and sustainability of these systems. This study aimed to analyze how variable-rate fertilization influences energy efficiency in coffee growing. A production area subjected to variable and fixed rates of fertilizer in alternating rows was evaluated. Geo-referenced yield data was collected to assess yield response for fixed and variable rate applications. The energy assessment was combined with the Geographic Information System (GIS) to determine site-specific energy indicators. To determine the energy flow, only NPK fertilizer applications were considered as inputs and the yield as output. The results obtained indicated that the variable rate fertilizer application has a small difference, indicating greater energy efficiency concerning the applied fertilizer and coffee production per crop season. It was observed in the 06/07 crop, the incorporated energy was 10.7 MJ kg$^{-1}$ for VR and 10.2 MJ kg$^{-1}$ for UR and for the 07/08 crop it was 30.7 MJ kg$^{-1}$ for VR and 34.9 MJ kg$^{-1}$ for UR. The energy balance was more efficient at variable rates, as it provided fertilizer savings without compromising yield. However, the difference between the embodied energy per mass of coffee produced was very small compared to the fixed rate.

**Keywords:** *Coffea arabica* L.; precision agriculture; perennial crop; energy balance

## 1. Introduction

Brazil is the world's largest coffee producer. *Coffea arabica* L. production occupies a total area of 1.5 million hectares, with an estimated production of 47.37 million bags of processed coffee (1 bag = 60 kg), and its largest producer is Minas Gerais state, with 3346 million bags of processed coffee [1].

Considering the potential yield and profit of coffee, a broad knowledge of high quality coffee production techniques is indispensable for modern coffee growing and new management tools such as precision agriculture should be used [2]. According to Mulla et al. (2010) [3] variability of soil attributes will influence the crop management efficiency and development. In this context, Precision Agriculture (PA) techniques offer management systems in which inputs are applied according to the spatial variability of the field using variable rate technology. PA's main objective is to provide economic and environmental benefits [4].

For some crops, such as grains, the study of techniques for monitoring the spatial variability of productivity is already consolidated, mainly due to harvesters having easier access to technological packages, unlike perennial crops such as orange, eucalyptus, and coffee [5,6].

Some authors have reported the effects of PA on input consumption and productivity indicating positive gains [1,7–10]. On the other hand, other assessments [11,12] also have suggested that different site-specific nutrient strategies were often not economically beneficial.

Furthermore, little is known about the actual environmental impact of this technology. Studies that perform economic and environmental analyses to test PA technologies [6,13–15] are rarer than those restricted to harvest evaluation. The determination of energy efficiency is an important tool when one wants to assess the sustainability of production systems [16]. Embodied energy indicates how demanding a production system is in terms of energy. Nutrient balances assess the balance (or not) between the input and output of nutrients into the system and, similarly, inform about the risk of nutrient loss and environmental contamination due to over-application [17,18].

In energy terms, an agricultural production system can be interpreted as a converter of solar energy into food energy, with the intervention of water, carbon dioxide and semi-manufactured products, such as fuels, fertilizers, pesticides, seeds, among others [19].

Input and output energy flows based on the material flows and production, respectively, can be used to assess the environmental sustainability of different management strategies [16,20,21] and nutrients [17,22]. Through PA adoption, data of yield and applied inputs are geographically available, allowing for site-specific economic and environmental indicators to be related to the field variability. For example, studies approaching energy analyses in citrus [6,23] and wheat crops [15] with site-specific variable fertilizer application provide maps of energy balance, embodied energy, and energy embodiment. The approach allowed for environmental management and sustainability analysis at the site-specific scale.

In coffee production systems, few studies have applied energy assessment, as highlighted by Lacerda et al. (2014) [24], Muner et al. (2015) [25] and Turco et al. (2018) [26] who applied the methodology to evaluate different coffee production systems. However, none of them evaluated variable rate fertilizer application.

This study aimed to analyze how variable rate fertilization influences energy efficiency in the coffee growing at one coffee plantation.

## 2. Materials and Methods

### 2.1. Coffee Areas and Treatments

Relevant data on the experimental design used for the present study are provided in this section and were obtained from a coffee farm located in Patrocínio municipality, Minas Gerais state, Brazil (18°41′31.14″ S, 46°49′18.17″ W), as it is referred to in the article of Molin et al., 2009 [27]. The studied plot had 4.67 ha, presenting 4-m distance between rows, cultivated with the species *Coffea arabica* L., Catuaí variety, conducted during two harvest seasons. The soil type was Yellow Oxisol [28]. According to Köppen classification, the climate is classified as Cwa.

A field-scale experimental design was used in which variable rate (VR) and uniform rate (UR) treatments were implemented in interspersed bands of two lines across the entire area (Figure 1). The design allowed the data to be interpolated separately for each treatment, allowing for comparisons between treatments. The inputs evaluated were N, P, K, whose sources were ammonium nitrate, triple superphosphate, and potassium chloride.

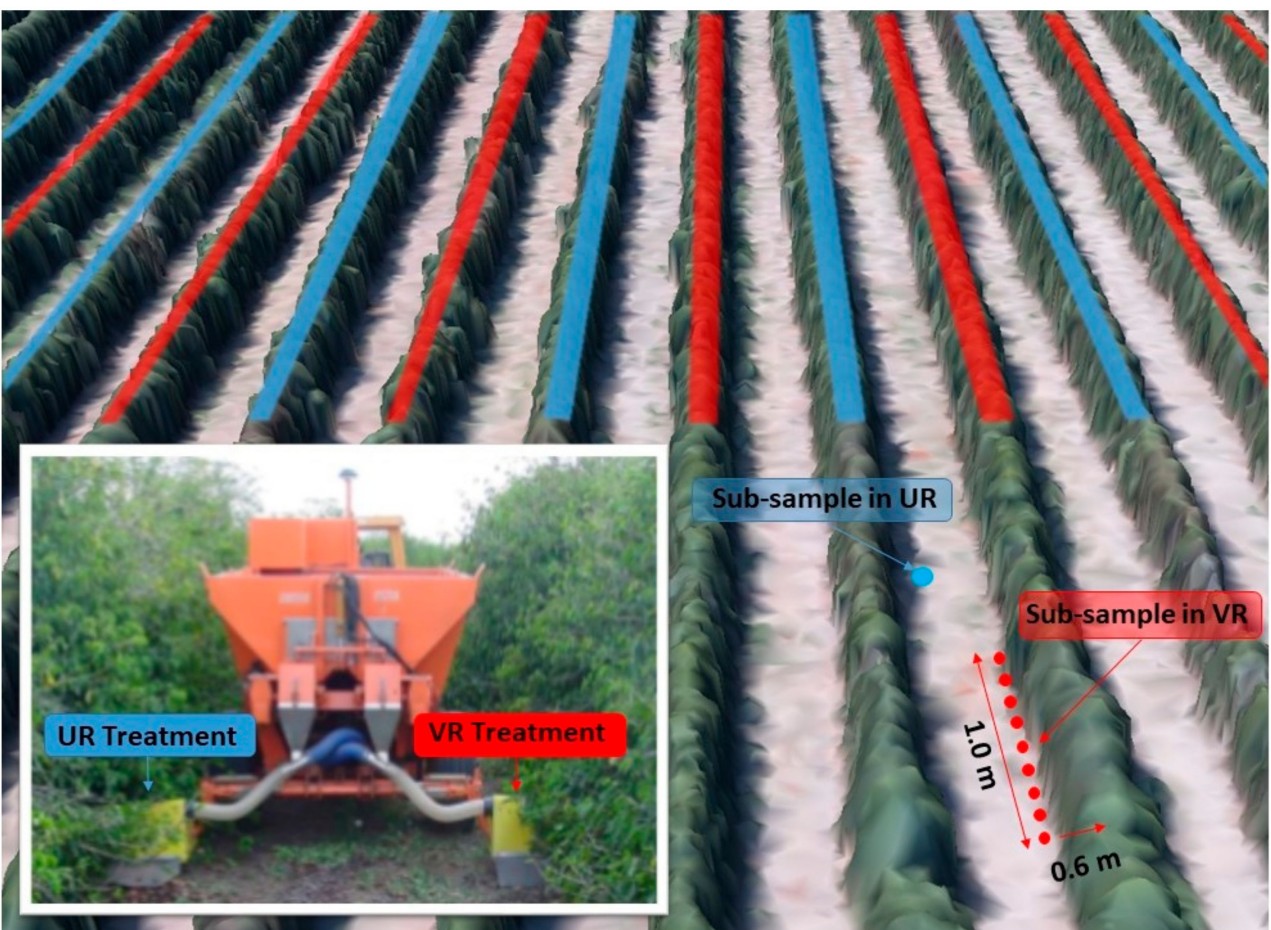

**Figure 1.** Example of the arrangement of treatments in each area.

For the VR, recommendations were made from soil sampling in the 0–0.2 m layer, the samplings were carried out at georeferenced points, and arranged in a grid with equidistant longitudinal and transverse spacing. For the soil samples in the UR treatment, we tried to respect the methodology used on the farms, where sub-samples are collected in the center of the lines along the area, composing a single sample composed by area [27].

In the VR treatment, each sample consisted of nine sub-samples, spaced parallel to the row of plants by 0.6 m and grouped at an interval of approximately 1.0 m (Figure 1).

For leaf sampling, collections were carried out only in the 2007/2008 season on branches located in the middle part of the plant height, the third and fourth pair of leaves, counted from the first pair of leaves with a length greater than 25.0 mm. Fifty-two leaves per sample were removed from thirteen plants close to the sampling point for each treatment using the average content in UR treatment. 2006/2007 season a uniform rate of 340.0 kg ha$^{-1}$ of urea was established based on the expected yield without sampling. The samples were placed in properly identified paper bags and stored in a Styrofoam container with ice to be taken to the laboratory.

The samples were sent to certified analysis laboratories to determine exchangeable contents of Phosphorus (P resin, mg dm$^{-3}$) and Potassium (K+, mmol$_c$ dm$^{-3}$) in the soil, and for the mineral nitrogen leaf content (N, g kg$^{-1}$), (Table 1).

**Table 1.** Statistical parameters of nitrogen, phosphorus and potassium analysis.

| Season | Treatment | Samples [1] | Mean | Min. | Max. | SD [2] | CV [3] (%) | W [4] |
|--------|-----------|-------------|------|------|------|--------|------------|-------|
| (leaf N, g kg$^{-1}$) | | | | | | | | |
| 07/08 | UR | 1 | 27.4 | - | - | - | - | - |
| | VR | 8 | 26.9 | 26.3 | 27.6 | 0.47 | 1.74 | 0.942 * |
| (soil P-resin, mg dm$^{-3}$) | | | | | | | | |
| 06/07 | UR | 1 | 18.0 | - | - | - | - | - |
| | VR | 20 | 17.9 | 12.0 | 35.0 | 7.1 | 39.3 | 0.791 |
| 07/08 | UR | 1 | 6.9 | - | - | - | - | - |
| | VR | 14 | 6.0 | 5.0 | 7.0 | 0.8 | 13.1 | 0.912 * |
| (soil K, mmol$_c$ dm$^{-3}$) | | | | | | | | |
| 06/07 | UR | 1 | 1.50 | - | - | - | - | - |
| | VR | 20 | 1.49 | 1.1 | 3.5 | 0.53 | 35.7 | 0.642 |
| 07/08 | UR | 1 | 4.50 | - | - | - | - | - |
| | VR | 19 | 4.47 | 3.7 | 5.0 | 0.43 | 9.6 | 0.778 |

[1] Number of samples considered after exploratory analysis; [2] Standard Deviation; [3] Coefficient of variation [4] Shapiro-Wilk Statistic; * Data came from a normal distribution at the 5% significance level.

For the maps, semivariograms of soil, leaves and yield data were modeled and adjusted (Table 2) and interpolated using the kriging method. R Studio software was used, with the stats [29], MASS [30] and geoR [31] packages.

The results obtained were used to determine the fertilizer application map by nutrient house regressions that met the recommended ranges and the expected yield. Fertilizer applications were conducted three separate times for nitrogen, twice for potassium, and with a single application for phosphorus. To establish the fertilizer doses that were applied, the expected yield (kg ha$^{-1}$) and the total length of plant rows (m) per hectare were considered. Furthermore, according to the results of the soil analysis, some adjustments were made to the result of phosphorus and potassium doses that were applied [27].

For fertilizer application, a drag fertilizer with a system by conveyors and gate (volumetric) and pneumatic directed distribution to the plant canopy projection, equipped with implementing unit for VR, GNSS receiver (C/A L1 carrier) without correction (Jacto S.A., model not yet commercially available) and pulled by a 31.7 kW tractor. Due to its construction, with two metering mechanisms and independent lateral distribution, it was possible to configure the left side of the machine to apply at a uniform rate and the right side to apply at a variable rate, following the recommendation by the application map previously defined and loaded into the computer of the equipment. The application map provides the equipment with information on the doses to be applied and their geographic locations.

For VR recommendations, fertilizer rates were calculated for each $10 \times 10$ m map pixel based on information from a soil fertility (or leaf nutrition) map and a yield map. The data comprised a period of two agricultural harvest seasons.

The doses for UR to be applied were determined considering expected yield ranges and levels found in the soil analysis report [2].

**Table 2.** Geostatistical parameters of nitrogen, phosphorus, potassium, and yield in the VR treatment.

| Season | Model | $C_0$ [1] | $C_1$ [2] | a (m) [3] | SC% [4] |
|--------|-------|-----------|-----------|-----------|---------|
| | | Nitrogen (leaf sample) | | | |
| 07/08 | Spherical | 0.05 | 0.21 | 140.0 | 19 *** |
| | | Phosphorus (soil sample) | | | |
| 06/07 | Spherical | 0.53 | 0.91 | 109.1 | 37 ** |
| 07/08 | Spherical | 0.003 | 0.001 | 105.2 | 75 ** |
| | | Potassium (soil sample) | | | |
| 06/07 | Spherical | 0.002 | 0.01 | 124.7 | 17 *** |
| 07/08 | Spherical | 2680.92 | 20,777.16 | 105.2 | 11 *** |
| | | Yield | | | |
| 06/07 | Exponential | $7.9 \times 10^3$ | $4.7 \times 10^3$ | 69.1 | 62.4 ** |
| 07/08 | Exponential | $2.1 \times 10^{-1}$ | $1.8 \times 10^{-1}$ | 57.3 | 53.9 ** |

[1] nugget effect; [2] structural variance; [3] range in meters; [4] Spatial Dependency (% Nugget). Note: For [32]: ** moderate spatial dependence; *** strong spatial dependence.

Linear regressions were established for each nutrient that met the recommended ranges and the expected yield, as presented by Molin et al. (2009) [27]. Recommendations were divided into three times for Nitrogen, twice for Potassium, and a single application for Phosphorus. The fertilizations used Urea (N source), Simple Superphosphate ($P_2O_5$) and Potassium Chloride ($K_2O$).

### 2.2. Material Flow and Energy Analysis

The embodied energy was determined based on Romanelli et al. (2010) [33], where the total input energy for fertilization was divided by yield (MJ kg$^{-1}$). This indicator represents the energy demand by fertilization to produce 1 kg of coffee beans. Input fertilization energy was determined by multiplying the application rates of nutrients (N, $P_2O_5$ and $K_2O$) by the energy indices. These indices represent the amount of energy needed to produce one unit of each input.

In this study, only the fertilizers that were used in each treatment were evaluated. The energy index used for each fertilizer was extracted from Pellizzi (1992) [34] for nitrogen (N) 74.0 MJ kg$^{-1}$ phosphorus ($P_2O_5$) 12.6 MJ kg$^{-1}$ and potassium ($K_2O$) 6.7 MJ kg$^{-1}$.

Output data were obtained by a self-propelled harvester Jacto, model K3 (Máquinas Agrícolas Jacto S.A.), equipped with a volumetric yield monitor, which measures the productivity in liters per hectare of the harvested coffee fruit [35]. To determine the yield during the experiment, samples of the coffee fruits were collected at the discharge outlet of the harvester. These samples were randomly obtained in the area, after they were sun dried for further processing and weighing, to generate a conversion factor, which was applied to the productivity data (green coffee) (L ha$^{-1}$) and made it possible to determine the productivity (kg ha$^{-1}$) of the processed coffee (commodity). For energy output determination, the energy index for coffee was adopted from [36] who determined 9.72 MJ kg$^{-1}$.

As noted by Colaço et al. (2020) [6], the energy incorporated in the mechanical operation was relatively less relevant when compared to the energy incorporated in the fertilizer itself. Thus, results depended mainly on the amount of fertilizer applied and the resulting yield for each year and treatment.

Energy balance (EB, Equation (1)), energy return on investment (EROI, Equation (2)) and energy intensity (EI, Equation (3)) were the indicators of energy efficiency. These indicators represent the amount of energy that was made available for each kg of coffee

produced. The incorporation of energy (Equations (1) and (2)) refers to the energy required to produce each unit of mass of processed coffee.

$$EB = OE - IE \tag{1}$$

$$EROI\_ = IE/OE \tag{2}$$

$$EI = IE/Productivity \tag{3}$$

where: EB—energy balance (MJ ha$^{-1}$); IE—input energy (MJ ha$^{-1}$) OE—output energy (MJ ha$^{-1}$); EROI is energy return on investment (non-dimensional), EI is–energy intensity (MJ ha$^{-1}$), Productivity (kg ha$^{-1}$).

*2.3. Spatial Variability Analysis*

Yield maps for the UR and VR treatments and fertilizer application maps (for the VR treatment) were generated for the entire area by kriging spatial interpolation, using in each case only the data from the corresponding ranges. Productivity and application maps were combined in layers and energy flows were determined ($10 \times 10$-m map pixel). Maps were produced using QGIS software (version 2.10; QGIS Development Team, 2019) [37].

## 3. Results

*3.1. Material Flow*

By strictly implementing the official fertilizer recommendation and using more detailed input information, VR reduced fertilizer application in both seasons mainly for potassium and phosphorus, yield varied between seasons, as coffee presents biennially. It is highlighted that the first crop had high productivity and the second one, low. Regarding material flows, variable rate provided a reduction of input consumption of the phosphate and potassium when compared with the conventional fertilization. The VR fertilization resulted in savings of 24.0% and 17.4% on P$_2$O$_5$ and K$_2$O respectively, in 06/07 season, and 17.4% and 20.9% in 07/08 season, for N there was no reduction (Table 3).

**Table 3.** Applied average doses and yield on coffee in seasons for uniform and variable fertilizer application rates.

| Season | Treatment | N (kg ha$^{-1}$) | P$_2$O$_5$ (kg ha$^{-1}$) | K$_2$O (kg ha$^{-1}$) | Yield (kg ha$^{-1}$) |
|---|---|---|---|---|---|
| 06/07 | VR | 340 | $13.7 \pm 4.4$ | $427.6 \pm 25.6$ | $2753 \pm 507$ |
|  | UR | 340 | 18.0 | 431.3 | $2835 \pm 383$ |
| 07/08 | VR | $274 \pm 56.4$ | $7.1 \pm 4.5$ | $104.3 \pm 45.5$ | $702 \pm 152$ |
|  | UR | 278 | 8.6 | 262.3 | $655 \pm 100$ |

Figure 2 shows the inputs of fertilizers, as it is possible to observe in the 2006/2007 season there was no variation in the application of nitrogen in the variable rate treatment and application of phosphorus and potassium occurred more uniformly. However, in the 2007/2008 season the greatest range occurred for all fertilizers in the variable rate treatment.

*3.2. Energy Flows*

The input energy was 28.2 GJ ha$^{-1}$ for VR and 28.3 GJ ha$^{-1}$ for UR in season 06/07 and 22.2 GJ ha$^{-1}$ for VR, 22.4 GJ ha$^{-1}$ for UR in season 07/08. When considering average yield, it was observed in season 06/07 that the difference between treatments was 82 kg ha$^{-1}$ (2.9%), which resulted in a very similar incorporation of energy between the two treatments, 10.7 MJ kg$^{-1}$ for VR and 10.2 MJ kg$^{-1}$ for UR. However, for 07/08 harvest season, it was 30.7 MJ kg$^{-1}$ for VR and 34.9 MJ kg$^{-1}$ for UR (Tables 4 and 5).

**Table 4.** Inputs and outputs used for energy calculation according to area, harvest, and treatment.

| Season | Treatment | N (MJ kg$^{-1}$) | | | P$_2$O$_5$ (MJ kg$^{-1}$) | | | K$_2$O (MJ kg$^{-1}$) | | |
|---|---|---|---|---|---|---|---|---|---|---|
| | | Max | Min | Mean $\pm$ s | Max | Min | Mean $\pm$ s | Max | Min | Mean $\pm$ s |
| 06/07 | VR | 14.14 | 6.7 | $9.09 \pm 1.4$ | 0.12 | $8.12 \times 10^{-8}$ | $0.06 \pm 0.02$ | 1.84 | 0.71 | $1.08 \pm 0.24$ |
| | UR | 14.14 | 6.7 | $9.05 \pm 1.3$ | 1.84 | 0.061 | $0.08 \pm 0.012$ | 1.62 | 0.78 | $1.03 \pm 0.15$ |
| 07/08 | VR | 53.8 | 19.3 | $29.6 \pm 5.1$ | 0.5 | $2.32 \times 10^{-7}$ | $0.12 \pm 0.08$ | 2.58 | 0.34 | $1.0 \pm 0.4$ |
| | UR | 43.8 | 19.6 | $32.1 \pm 4.4$ | 0.23 | 0.1 | $0.16 \pm 0.02$ | 3.74 | 1.67 | $2.7 \pm 0.4$ |

**Table 5.** Energy summary of NPK fertilizers and yield for each treatment, area, and crop. Difference between treatments and % of the difference between treatments.

| Season | Treatment | Fertilizers (MJ kg$^{-1}$) | Δ (MJ) | Δ (%) |
|---|---|---|---|---|
| 06/07 | VR | $10.7 \pm 2.2$ | $-0.5$ | $-4.6$ |
| | UR | $10.2 \pm 1.5$ | | |
| 07/08 | VR | $30.7 \pm 5.3$ | 4.2 | 12 |
| | UR | $34.9 \pm 4.8$ | | |

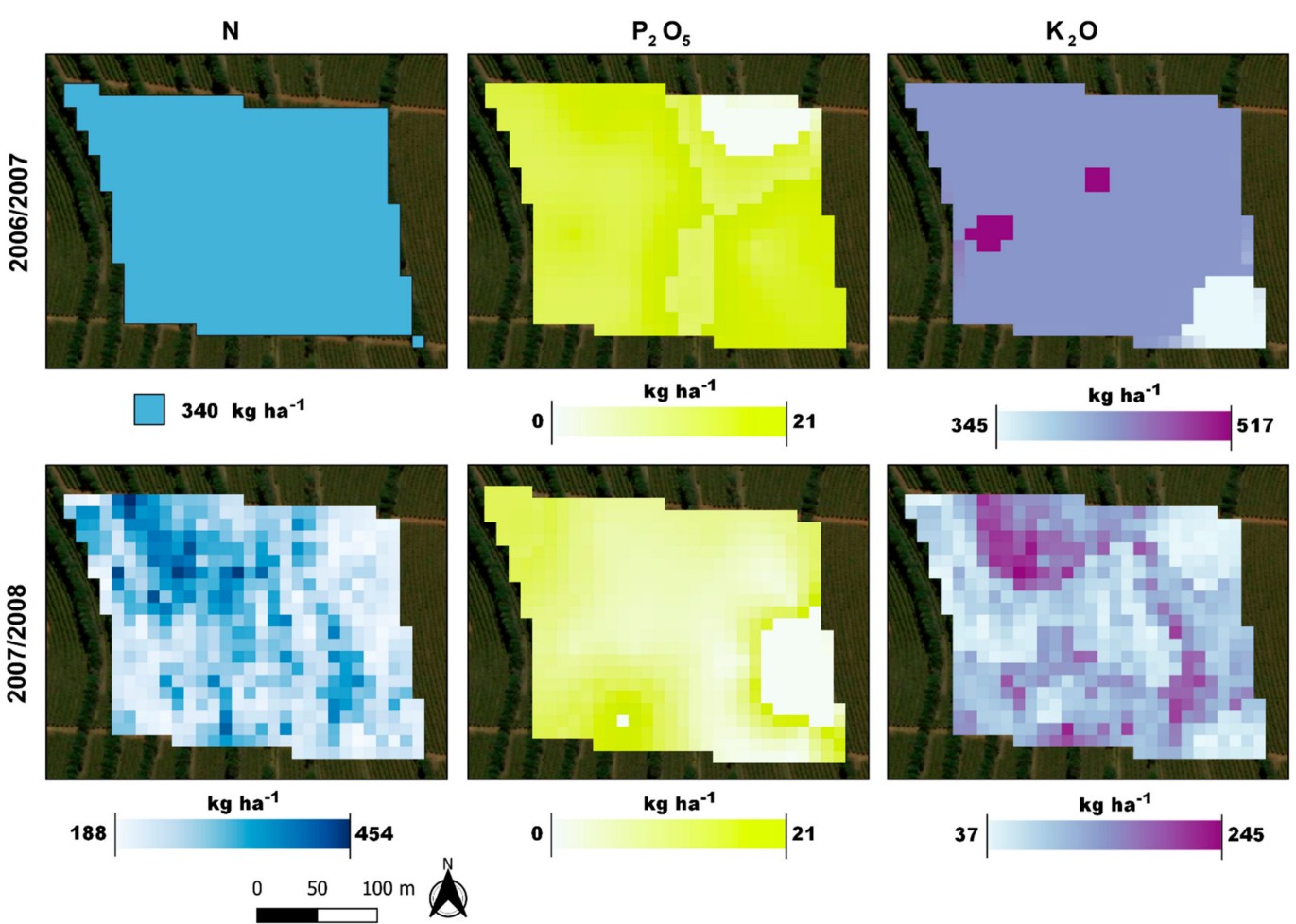

**Figure 2.** Applied fertilizer doses in seasons 2006/2007 and 2007/2008.

The maximum value energy embodied by nitrogen fertilizer in every kilo of coffee in VR was 14.1 MJ kg$^{-1}$ and minimum was 6.7 MJ kg$^{-1}$ in season 06/07. For season 07/08 the maximum value was 53.8 MJ kg$^{-1}$ and 19.3 MJ kg$^{-1}$ for minimum value (Table 4). Nitrogen accounted for 85% of the energy incorporated by fertilizers.

Implementing the official fertilizer recommendation and using more detailed input information, VR reduced overall fertilizer application in both seasons and the differences between the spatial patterns of each treatment for embodied energy maps by fertilizer are shown in Figure 3.

The energy indicators are presented in Table 6. Energy efficiency in this study was determined by measuring the net energy from the fertilization of the arabica coffee. The comparison of the net energy of the fertilization in VR and UR arabica coffee was −0.98 MJ kg$^{-1}$ coffee in VR treatment and −0.48 MJ kg$^{-1}$ coffee in season 06/07; however, in season 07/08 the yield was low and net energy was the 20.98 MJ kg$^{-1}$ coffee in VR and 25.18 MJ kg$^{-1}$ coffee in UR. EROI was higher for UR in 06/07 and for VR in 07/08. The energy incorporated (EI) was similar for both in 06/07, and slightly lower for VR in 07/08.

**Table 6.** Energy performance of evaluated treatments.

| Season | Treatment | EB | EROI | EI MJ kg$^{-1}$ |
|---|---|---|---|---|
| 06/07 | VR | −0.98 | 0.90 | 10.7 |
| | UR | −0.48 | 0.95 | 10.2 |
| 07/08 | VR | −20.98 | 0.31 | 30.7 |
| | UR | −25.18 | 0.28 | 34.9 |

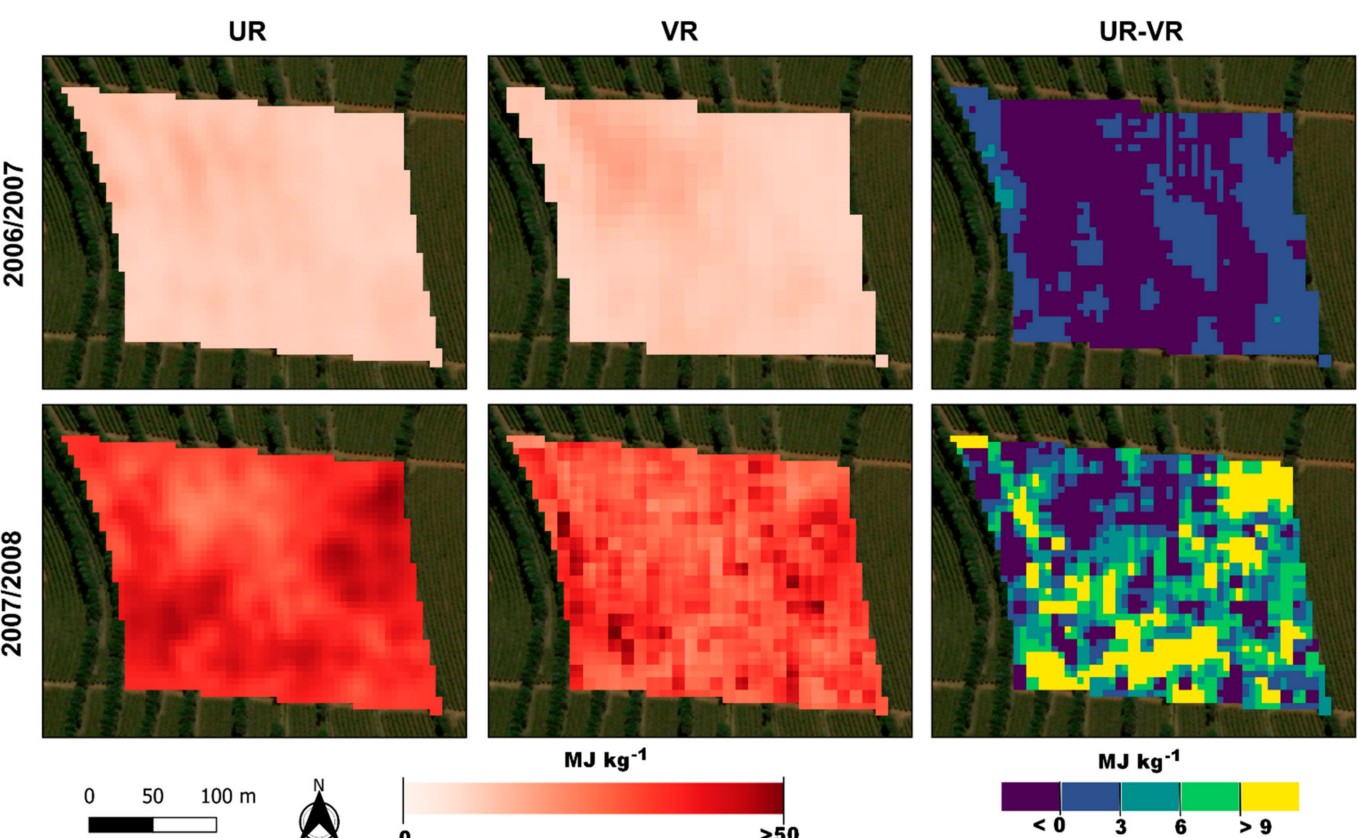

**Figure 3.** Maps of embodied fertilizer energy for UR, VR, and the difference between treatments (VR-UR) over two seasons.

## 4. Discussion

Studies that have attempted to measure the economic benefits of variable rate over traditional management typically test the claim that variable rate provides input savings without compromising yield, or that using the same amount of inputs increases yield [38]. Although there is a minute difference in incorporated energy between treatments in season 06/07 (Table 5), our results also point to energy gains when using the variable rate, be-sides observing a reduction in the amount of fertilizer.

The biggest difference was in the 07/08 harvest, obtaining an energy reduction value of 4.2 MJ kg$^{-1}$ of coffee. So, variable rate application presented a reduction in input energy and an increase in the output energy availability, that is, productivity.

The variable rate application of fertilizer on coffee was evaluated by Molin et al. (2010) [2] and they observed a similar trend, with the VR providing 23% savings in phosphorus and 13% increase in potassium consumption when compared to UR. Regarding productivity, they observed that plots that received fertilizers at VR showed a productivity increase of 34% compared to UR. In our study we observed a productivity increase of 7% only in season 07/08. We observed a reduction for VR in the use of all fertilizers in the 07/08 crop, with a reduction of 21% for K$_2$O and 17% for P$_2$O$_5$ and in the 06/07 crop, the reduction in the use of phosphorus was 24%.

When evaluating the energy demand in the orange crop, Colaço et al. (2020) [6] observed that nitrogen was the most demanding input in terms of energy, accounting for 66% and 73% of the total fertilization energy incorporated in treatments VR and UR of orchard 1, and for 56% and 74% in treatments VR and UR of orchard 2.

In our study, nitrogen, as the main energy input, represented 89% of the energy incorporated by fertilizers. The magnitude of N for the energy input was expected, since N has the highest energy index among the evaluated inputs, that is, it requires more energy to be produced than other inputs. As observed through the results in the 06/07 season the energy incorporation between variable and fixed rates showed a small difference, being smaller in the fixed rate since fertilization of nitrogen was the same value in both treatments and the yield of the 10 kg ha$^{-1}$ was larger in UR. In the 07/08 season, the amount of nitrogen applied was different and resulted in a difference of 4.2 MJ kg$^{-1}$ of incorporated energy between the two treatments.

Despite the fact that greater nutrient surpluses cannot necessarily be interpreted as resulting from excessive nutrient application because coffee is perennial, this can, however, indicate that nutrient losses were more likely to occur in the UR treatment than in the VR, as we observed a reduction in excess phosphorus and potassium in both seasons in the VR. VR treatment reduced excess P$_2$O$_5$ in both seasons.

Other studies have evaluated the energy sustainability arabica coffee production systems in Brazil. In Muner et al. (2015) [25] three production systems were evaluated, and the results showed that energy intake related to fertilizer inputs had statistical differences among the three cropping systems. Cultivation systems with good practices showed higher energy consumption, that such systems had higher fertilizer consumption, followed by conventional systems and organic coffee farms. The reduced contribution regarding fertilizer inputs of organic coffee farms was justified by its distinctive nutritional management, which adopted organic fertilization as the base of plant nutrition. These authors, highlighted that the highest energy intake of conventional systems (65.9%) was due to an intensive use of fertilizers, which is widely applied in conventional systems. Thus, the embodied energy was 12.4 MJ kg$^{-1}$ for conventional systems, 4 MJ kg$^{-1}$ for organic coffee farms, and 14.3 MJ kg$^{-1}$ for cultivation systems with good practices. In our study the embodied energy was between 10 to 30.2 MJ kg$^{-1}$ for a year with high productivity and a year with low productivity, respectively.

So, another aspect that should be noted is productivity, as coffee presents a biannual production [39] which is possible to observe in the result in energy embodied (Tables 4 and 5). This aspect resulted in a 3-fold increase in the embodied energy per kg of coffee in the low production season.

The energy intensity (EROI) was higher in the VR treatment in the 07/08 season. This indicates that the VR is more energy-efficient because it incorporates less per kg of coffee produced. Data with the same trend are found by Colaço et al. (2012) [15] when they evaluated the VR treatment in wheat and obtained higher EROI values compared to the values obtained with the UR treatment. [40] estimated Energy Balance and Green House Gas Emission on Smallholder Java Coffee Production of Indonesia, the results showed that the energy value obtained for the arabica coffee transportation and production were 152.36 MJ t$^{-1}$ and 74.54 MJ t$^{-1}$, respectively, while the robusta coffee had an energy value of 158.68 MJ t$^{-1}$ and 162.74. MJ t$^{-1}$, respectively. These authors observed a net energy value positive of 9.1 and 8.8 for arabica and robusta coffee respectively, different from the values observed in our study.

## 5. Conclusions

The analysis of an activity through the lens of energy is an important tool for observing production systems that use precision agriculture as a management tool, as it can provide subsidies to monitor the energy potential of crops. The energy indicators pointed to precision agriculture through variable rate as a possible tool capable of improving energy sustainability in a coffee production system. Embodied fertilization energy was reduced by approximately 4.2 MJ kg$^{-1}$ coffee in a season with a nitrogen rate variable. The VR fertilization resulted on savings of 24.0% and 17.4% on $P_2O_5$ and $K_2O$ respectively, in the 06/07 season, and 17.4% and 20.9% in the 07/08 season.

As the difference between the results was small, it is suggested that future studies work with a larger number of crops and areas. In addition, for future studies, it is suggested that the energy assessment takes into account the entire production cycle, allowing the total balance of the activity to be observed.

**Author Contributions:** Conceptualization, G.A. and M.M.; methodology, G.A., M.M. and T.L.R.; software, M.M.; data curation, G.D.C.F. and J.P.M.; writing—original draft preparation, G.A. and M.M.; writing—review and editing, G.A., M.M., T.L.R., G.D.C.F. and J.P.M.; supervision, T.L.R. All authors have read and agreed to the published version of the manuscript.

**Funding:** This research received no external funding.

**Institutional Review Board Statement:** Not applicable.

**Informed Consent Statement:** Not applicable.

**Conflicts of Interest:** The authors declare no conflict of interest.

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
