# Peer review of "Energy Efficiency of Variable Rate Fertilizer Application in Coffee Production in Brazil"

_agriengineering, doi:10.3390/agriengineering3040051_

Round 1

Reviewer 1 Report

This manuscript is well written and easy to follow. The authors tried to explain the energy efficiency of variable rate fertilizer application in coffee production. Out of the two seasons evaluated one season does not have any variation in N fertilizer for Uniform Rate and Variable rate. It is obvious that less fertilizer application requires less energy.  With both VR and UR rates producing comparable yields, VR rates should be more energy-efficient as shown in session 07/08 about 12% with 4.2MJ/kg. I do not see much novelty in the work. However, the article describes the energy utilized per kg of yield produced for both UR and VR rates specific to Coffee production in Brazil.

Specific comments:

Line 154: Check equation 2 and its definitions for OIE and IOE

Author Response

We thank Reviewer 1 for the comments provided. The equations were adjusted.

Although the methodology is known, its use on perennial crops has not been so widely used. A 2-year evaluation may bring the framework for further long-term analysis. The environmental benefits of tools such as Precision Agriculture should be more explored, since low energy and, similarly, low carbon economies have been desired. 

Reviewer 2 Report

Interesting work, generally well prepared. However, it needs more corrections, especially the description of the methodology. Requires english linguistic correction.

Abstract - requires linguistic correction.

Introduction written correctly.

Line 44-45: “For some crops, such as grains, the study of the spatial variability of yield is already consolidated, mainly due to the easier access to technological packages that make this in vestigation possible.”

The authors probably meant that technical solutions for yield mapping in cereals (grains) are available. What else is " already consolidated". I suggest rephrasing the sentence to make this more explicit.

Material and methods

Line 80:  „in a commercial area” .  Requires rewording.

Line 92: The recommendations for VR were made from soil sampling in the 0-0.2 m layer carried out by [27].

It absolutely must be rewritten by who?  I propose "... "performed and described in Faulin G.C (27)". . The publication is not in English. Some most important information about soil sampling, methods for determination of N, P and K should be added. How were the samples taken? By hand? Quad? How were the samples analysed? What parameters were analysed? N mineral in soil also? By what methods?

Line 98: „For fertilizer application, a drag fertilizer with a volumetric meter of two independent conveyors with adjustable gates was used, equipped with variable rate”

State the model and manufacturer of the fertiliser spreader.

„equipped with variable rate” – Equipped with implementing units for VR rather than variable rate. What VRA adjustment system was used? By weight? Change in shutter geometry? Other?

„and Global 99 Navigation Satellite System (GNSS)” – equipped with a GNSS receiver, not GNSS technology.

Specify receiver model and positioning accuracy (with correction without correction, with what correction?)

What terminal was used? Manufacturer required. From the spreader? From the tractor?

Lina 107: How were the application maps prepared? What interpolation methods were used?  What software? Was the same method used as for the yield maps?

Liniea 108: "in the leaf (for N) or soil (for P and K);" Why "or"? Which map was prepared using soil data and which using leaf data?

How 'leaf data' was collected. Sensor?

Linia 108: „in general, fertilizer 108 rates were higher for lower nutrient levels and for higher yield expectationsÄ™

Be precise in stating which fertilisation strategy was used. Qualitative? Yield levelling?

Linia 111 “The data comprised a period of two agricultural harvest seasons, between 2006 and 2008. Which two seasons 2006? 2007? 2008?

The chapter is missing a methodology for determining 'Leaf N content (g kg-1) “

Linia 123: Table 1.

P - is this exchangeable potassium? Total? Other?

K - is potassium exchangeable? Total? Other?

N - mineral nitrogen? Total?

Line 137: „as described by” By who?

Line 137” Output data were obtained as described by [32].” The publication is 20 years old. Requires clarification. 

Linie 153: Equations 1-3 are not legible and are not written correctly. Require rewording.

Line 164: „by spatial interpolation” What interpolation method? Kriging? Another?

Did the yield mapping take into account the harvesting delay between the harvest team and the leaf measurement site?

Wyniki

No soil maps or soil analysis results available. No information on variability of soil parameters. (Not content as in Table 1, but with spatial variability)

Figure 2. requires major editorial correction. (e.g. P2O5)

Table 3. „ mean 9.09±1.4 „ What does ± mean when max and min are given?

Discussion and Conclusions.

No major objections

Author Response

Authors would like to thank the comments made by Reviewer 2, which helped us to improve the manuscript.

Below, you may find comments over the specifics comments.

Abstract - requires linguistic correction.

Reply: was adjusted.

Introduction written correctly.

Line 44-45: “For some crops, such as grains, the study of the spatial variability of yield is already consolidated, mainly due to the easier access to technological packages that make this in vestigation possible.”

The authors probably meant that technical solutions for yield mapping in cereals (grains) are available. What else is " already consolidated". I suggest rephrasing the sentence to make this more explicit.

 Reply: was adjusted.

Material and methods

Line 80:  „in a commercial area” . Requires rewording.

Reply: was adjusted.

Line 92: The recommendations for VR were made from soil sampling in the 0-0.2 m layer carried out by [27].

It absolutely must be rewritten by who?  I propose "... "performed and described in Faulin G.C (27)". . The publication is not in English. Some most important information about soil sampling, methods for determination of N, P and K should be added. How were the samples taken? By hand? Quad? How were the samples analysed? What parameters were analysed? N mineral in soil also? By what methods?

Reply: this part has been rewritten and added more details about sampling in line 96-114. 

Line 98: „For fertilizer application, a drag fertilizer with a volumetric meter of two independent conveyors with adjustable gates was used, equipped with variable rate”

State the model and manufacturer of the fertiliser spreader.

Reply: insert in lines 1331 - 137.

„equipped with variable rate” – Equipped with implementing units for VR rather than variable rate. What VRA adjustment system was used? By weight? Change in shutter geometry? Other?

Reply: was adjusted.

“and Global 99 Navigation Satellite System (GNSS)” – equipped with a GNSS receiver, not GNSS technology.

Reply: was adjusted.

Specify receiver model and positioning accuracy (with correction without correction, with what correction?)What terminal was used? Manufacturer required. From the spreader? From the tractor?

Reply: Details was inserted in text

Lina 107: How were the application maps prepared? What interpolation methods were used?  What software? Was the same method used as for the yield maps? soil data and which using leaf data? How 'leaf data' was collected. Sensor?

Reply: Details was insert in line 121-123

Liniea 108: "in the leaf (for N) or soil (for P and K);" Why "or"? Which map was prepared using

Linia 108: „in general, fertilizer 108 rates were higher for lower nutrient levels and for higher yield expectationsÄ™

Be precise in stating which fertilisation strategy was used. Qualitative? Yield levelling?

Reply: was adjusted.

Linia 111 “The data comprised a period of two agricultural harvest seasons, between 2006 and 2008. Which two seasons 2006? 2007? 2008?

Reply: was adjusted.

The chapter is missing a methodology for determining 'Leaf N content (g kg-1) “

Reply: Details was inserted in text.

Linia 123: Table 1.

P - is this exchangeable potassium? Total? Other?

K - is potassium exchangeable? Total? Other?

N - mineral nitrogen? Total?

Reply: was adjusted.

Line 137: „as described by” By who?

Reply: was adjusted.

Line 137” Output data were obtained as described by [32].” The publication is 20 years old. Requires clarification. 

Reply: the harvester was described

Linie 153: Equations 1-3 are not legible and are not written correctly. Require rewording.

Reply: was adjusted.

Line 164: „by spatial interpolation” What interpolation method? Kriging? Another?

Reply: was adjusted.

Did the yield mapping take into account the harvesting delay between the harvest team and the leaf measurement site?

 Reply: The adjustment was carried out taking into account the harvester's delay in order to obtain the real position of plant yield.

Wyniki

No soil maps or soil analysis results available. No information on variability of soil parameters. (Not content as in Table 1, but with spatial variability)

Reply: The statistical and geostatistical parameters was described (Tables 1 and 2)

Figure 2. requires major editorial correction. (e.g. P2O5)

Reply: The P2O5 was adjusted.

Table 3. „ mean 9.09±1.4 „ What does ± mean when max and min are given?

Reply: Was insert standard deviation symbol.

Round 2

Reviewer 2 Report

The authors made the needed changes to the manuscript, the most relevant.

Minor corrections and suggestions remain:

Line 109: specify what forms of P, K and N were determined in the laboratory, P - is this exchangeable potassium? Total? Other? K - is potassium exchangeable? Total? Other? N - mineral nitrogen? Total?

Line 130: “GPS reciver” change to: “GNSS reciver”

Line 161: Is „volumetric harvesting monitor”. Change to “volumetric yield monitor”.

Line 162: “coffee fruit.as” >>”coffee fruit as”

Line 212: Figure 2 - UR and VR are mentioned in the description. There is no UR in the graphics, except for N. Correction required.

Author Response

Dear Reviewer,

Thank you for the comments and suggestion made, which allowed us to improve the manuscript. Please, find below, the actions made for each specific comment.

Line 109: specify what forms of P, K and N were determined in the laboratory, P - is this exchangeable potassium? Total? Other? K - is potassium exchangeable? Total? Other? N - mineral nitrogen? Total?

Line 109 was replaced by “The samples were sent to certified analysis laboratories to determine exchangeable contents of Phosphorus (P resin, mg dm-3) and Potassium (K+, mmolc dm-3) in the soil, and for the mineral nitrogen leaf content (N, g kg-1), (Table 1).”Line 130: “GPS reciver” change to: “GNSS reciver”

It was corrected

Line 161: Is „volumetric harvesting monitor”. Change to “volumetric yield monitor”.

It was corrected

Line 162: “coffee fruit.as” >>”coffee fruit as”

It was corrected

Line 212: Figure 2 - UR and VR are mentioned in the description. There is no UR in the graphics, except for N. Correction required.

It was corrected